# GC-MS/MS Determination of Steroid Hormones in Urine Using Solid-Phase Derivatization as an Alternative to Conventional Methods

**DOI:** 10.3390/molecules27185796

**Published:** 2022-09-07

**Authors:** Azamat Temerdashev, Pavel Nesterenko, Ekaterina Dmitrieva, Kseniya Zhurkina, Yu-Qi Feng

**Affiliations:** 1Analytical Chemistry Department, Kuban State University, Stavropolskaya st., 149, 350040 Krasnodar, Russia; 2Physical Chemistry Department, M.V. Lomonosov Moscow State University, Leninskie Gory, 1, 119991 Moscow, Russia; 3Department of Chemistry, Wuhan University, Wuhan 430072, China

**Keywords:** steroid hormones, androgens, estrogens, LLE, SPAD, GC-MS/MS, sample preparation

## Abstract

Solid-phase analytical derivatization (SPAD) is a promising hybrid sample preparation technique combining the clean-up and preconcentration of the sample in a single step. In this work, a novel SPAD method based on the preparation of trimethylsilyl (TMS) derivatives of steroid hormones (testosterone, estrone, DHT, estriol, estradiol, and progesterone) in Phenomenex Strata C18-E (100 mg, 1 mL) cartridges has been developed and applied for their GC-MS/MS determination in human urine samples. The proposed procedure allows the detection and quantification of steroids with limits of 1.0–2.5 and 2.5–5 ng/mL, respectively. These characteristics are comparable with those obtained with a conventional liquid–liquid extraction, while the recovery of analytes in the proposed SPAD procedure is higher. The major advantages of SPAD are a short derivatization time, high efficiency, and the possibility to automatize the procedure. However, its cost-effectiveness in routine practice is still questionable.

## 1. Introduction

Steroid hormones are endogenous compounds synthesized from cholesterol in gonads, placenta, and adrenal glands that play an important role in regulatory functions. They are present in the body at low concentrations, and their changes are sometimes associated with such diseases as depression, cancer, and even diabetes [1]. Steroids are also an important part of an athlete biological passport according to the World Anti-Doping Agency (WADA) regulations [2,3,4,5]. Thus, the accurate determination of steroids is considered an important task to prevent false results and explore the mechanism of diseases associated with steroid hormones.

Immunoassay (IA) and gas or liquid chromatography hyphenated with mass spectrometry are the major techniques used for steroid quantification. The main advantage of IA is an easy and quick implementation of the method in a laboratory with high sensitivity, but it has a substantial limitation connected with a single analyte measurement at a time and insufficient selectivity (in the case of “hook effect” occurrence) [6,7,8,9]. Therefore, high-performance liquid chromatography coupled with mass spectrometry (LC-MS) and gas chromatography-mass spectrometry (GC-MS), especially with triple quadrupole mass spectrometry (MS/MS), and high-resolution mass spectrometry (HRMS) techniques have become preferable [10,11,12,13,14,15,16,17,18,19]. The application of LC-MS(/MS) is complicated due to the presence of isobaric compounds, limited separation efficiency and selectivity of HPLC or UHPLC columns along with strong matrix effects in the case of electrospray ionization (ESI). GC-MS(/MS) detection allows a significantly higher separation efficiency and provides much more accurate results as compared to LC-MS(/MS), but the sample preparation for GC-MS detection is often cumbersome. Modern methods of chromatography and mass spectrometry combined with ion-mobility spectrometry (IMS) have opened a new era for their measurement with high sensitivity and specificity and have made extended steroidomic research possible [20,21,22].

Considering a wide list of potential analytes, nowadays, the most common sample preparation techniques are dilute-and-shoot, liquid–liquid extraction (LLE) [23,24,25], dispersive liquid–liquid microextraction (DLLME) [26,27,28,29], as well as solid-phase extraction (SPE) [30,31,32] coupled with mineral or enzymatic hydrolysis. As a rule, hydrolysis is applied in the case of urinalysis, since steroid hormones are mostly present in the conjugated form (as glucuronides or sulfates) in this matrix and cannot be detected directly by GC-MS(/MS). Moreover, the determination of conjugated steroid hormones does not allow the assessment of the total content of some steroids, e.g., testosterone, etiocholanolone, etc., having important diagnostic value in doping control and clinical diagnostics.

Since concentrations of steroid hormones in urine vary by several orders of magnitude (from units to hundreds of ng/mL), their quantification is often accompanied by a derivatization step after LLE or SPE concentration. For GC-MS(/MS) detection, a mixture of dithiothreitol (DTT), N-methyl-N-(trimethylsilyl)trifluoroacetamide (MSTFA), and NH_4_I is a commonly used derivatization reagent. This reagent allows the silylation of even enol groups thus resulting in an increase in the intensity of a molecular ion and consequently in higher sensitivity of analysis. In the case of HPLC determination of steroids, the use of derivatization reagents is less common; however, when their application is required, hydroxylamine can be used [33] as a simple and efficient reagent compatible with aqueous media.

Despite the efficiency of the above-described approaches, they have a few shortcomings, the foremost of which is a time-consuming sample preparation using qualified labor. In this way, the sample preparation consisting of LLE with subsequent derivatization for GC-MS(/MS) detection requires more than 120 min, and steroid recoveries are strongly dependent on the extractant and pH of the medium. In addition, such sample preparation has limited capabilities for automation.

The details and characteristics of some representative methods for the determination of steroids are presented in Table 1.

This cutting-edge research is aimed to provide new information about steroidome and discover new potential disease markers. However, the development of novel quick, easy, and efficient sample preparation methods for steroid determination is still an actual and important task in the routine practice. Solid-phase analytical derivatization (SPAD) is a promising sample preparation techniques. The main advantage of this method is the coupling of clean-up, preconcentration, and derivatization in one step. Some aspects of this method were previously described in [27,39,40], but a few blind spots still exist in this technique in comparison with conventional sample preparation methods.

The aim of this research was to investigate the possibilities of SPAD sample pretreatment for GC-MS/MS determination of trimethylsilyl (TMS) derivatives of steroid hormones in urine and compare its efficiency and selectivity with a conventional liquid–liquid extraction (LLE) sample preparation method.

## 2. Results

### 2.1. Optimization of Sample Preparation Conditions

To optimize derivatization conditions, first, the influence of the derivatization reagent concentration on the formation of trimethylsilyl-derivatives was evaluated. For this, 100 µL of either undiluted or 2-, 5-, 10-, 20-fold diluted derivatization reagent with acetonitrile was used to react with analytes on a cartridge. The results have revealed that an increase in the dilution factor of the derivatization reagent led to decreased analyte peak areas; therefore, 100 µL of the undiluted derivatization reagent was used in subsequent experiments.

Then, the temperature of the SPE cartridge during the solid-phase preparation of TMS derivatives was optimized. The temperature effect was studied at ambient temperature (22 °C), 40 °C, 50 °C, 60 °C, 70 °C, and 80 °C. Higher temperatures were not tested in order to avoid the possible deformation of SPE cartridge polymer parts. A temperature of 80 °C was selected as optimal for the future experiments as it provided the highest analyte responses.

The derivatization time was varied between 1 and 90 min to find the optimum. According to the obtained results, thermostating at 80 °C for 10 min was sufficient for the preparation of all TMS derivatives by using an undiluted derivatization reagent. The obtained yields of the derivatization reactions are similar for all steroids used in this work, as shown in Figure 1.

### 2.2. Chromatographic Separation and Mass Spectrometric Detection Conditions

The separation of testosterone, estrone, DHT, estriol, estradiol, and progesterone was obtained under the following temperature gradient program: an initial temperature of 150 °C was held constant for 2 min; then, it increased to 315 °C at the flow rate of 7 °C/min and was held constant for 25 min. Carrier gas (helium) flow rate was set at 1 mL/min, split ratio—1:5, septum purge—3 mL/min. A triple quadrupole mass spectrometer equipped with an electron ionization ion source operated in multiple reaction monitoring (MRM) mode. The optimized conditions of MRM detection are shown in Table 2. Collision gas (argon) pressure was 1.2 mTorr; cycle time—0.3 s.

The preparation of urine samples for analysis strongly affects the results of the quantification of steroids. Considering that optimum pH values in LLE extraction (Figure 2) are different for steroid hormones of different classes, their recoveries can be insufficient when unified LLE conditions are used. To overcome this drawback, the application of either SPE with hydrophobic cartridges or sample preparation under different LLE conditions can be considered. To achieve a higher sensitivity of GC-MS steroid detection, an extraction step is typically followed by derivatization using diverse reagents. Among the derivatization reactions, silylation is the most commonly used. The use of a derivatization reagent mixture, namely MSTFA/NH_4_I/DTT, allows the detection of highly abundant molecular ion peaks for analytes. However, it usually takes approximately 40 min to complete the reaction. As a result, the sample preparation includes a large number of time-consuming and tedious steps and can result in analyte losses.

To overcome these limitations, a combination of SPE with analytical derivatization can be used. A comparison of conventional procedures for steroid determination with SPAD is presented in Table 3.

### 2.3. Analytical Figures of Merit of the Proposed Method

Blank urine samples obtained according to the procedure described in Section 4.3. “Urine samples” were spiked with the selected analytes to obtain the final concentrations within the range of 1.0–250 ng/mL. The experiment was carried out under the optimized conditions; the internal standard (methyltestosterone) concentration was 50 ng/mL. Analytical figures of merit of the developed SPAD procedure are presented in Table 4. The detection limit (LOD) corresponded to the lowest detected analyte concentration with the signal-to-noise ratio of 3, while the quantification limit (LOQ) was determined at the signal-to-noise ratio of 10.

As can be seen from Table 5, the LOD and LOQ values are comparable for LLE and SPAD sample preparation procedures with similar conditions of MS detection, which makes the developed procedure suitable for doping control and clinical diagnostic purposes. Recovery values were calculated as the ratio of peak areas obtained by analyzing standard solutions of steroid hormone TMS derivatives to those of solutions that passed through the sample preparation with equal final concentrations of the analytes.

### 2.4. Analysis of Real Samples

The chromatograms of real male and female urine samples are shown in Figure 3 and Figure 4, illustrating the applicability of the proposed SPAD method to the analysis of real samples.

## 3. Discussion

In this study, a procedure containing the step of TMS-derivatives preparation is presented. According to the reaction conditions, the presence of moisture in the cartridge at the stage of the derivatization reagent loading should be avoided. This can be easily achieved for conventional procedures by solvent evaporation from tubes, but it is harder to control it in the case of the sorbent. The presence of the developed porous structure makes it complicated and requires excessive drying time to prevent hydrolysis of MSTFA. It should be noted that SPAD also requires the absence of residual silanols or other functional groups on the sorbent surface, which can react with a derivatization reagent, so the use of cartridges with end-capped alkyl silica is required. As a result, this methodology cannot be used for extremely polar compounds with this combination of the non-polar C18 sorbent and a derivatization agent such as MSTFA. At the same time, this limitation based on analytes retention could be used in further research as one of the instruments of the selectivity control.

The main disadvantage of the proposed SPAD method is its relatively high cost of analysis due to the use of SPE cartridges. This is an extremely important aspect for high-throughput laboratories performing routine analysis. However, the negative effect of it can be partly compensated for by the substantial decrease in sample preparation time [40] and the increased number of analytes that can be determined in a single run. 

## 4. Materials and Methods

### 4.1. Chemicals

Standards of testosterone (T), methyltestosterone (MT), dihydrotestosterone (DHT), estrone (E1), 17α-estradiol (17α-E), estriol (E2), progesterone (P), N-methyl-N-(trimethylsilyl)trifluoroacetamide (MSTFA), dithiothreitol (DTT), and NH_4_I were from Sigma-Aldrich (St. Louis, MO, USA); β-glucuronidase from *Escherichia coli* (*E. coli*) was from Roche Diagnostics (Mannheim, Germany). Dipotassium hydrogen phosphate, disodium hydrogen phosphate dihydrate, and sodium azide (all 99%) used for the preparation of a phosphate buffer solution (pH 6.5) were obtained from Vecton (St. Petersburg, Russia). HPLC-grade acetonitrile was obtained from Biosolve (Jerusalem, Israel).

### 4.2. Preparation of Solutions

Standard solutions of the analytes (17α-estradiol, estriol, estrone, testosterone, dihydrotestosterone, and progesterone) with a concentration of 1 mg/mL were prepared in methanol. They were diluted in methanol to obtain calibration solutions within the range of 1.0–250 ng/mL. Methyltestosterone was used as the internal standard with a concentration of 50 ng/mL in the final sample. Calibration solutions were stored at 4 °C; stock solutions were stored at −20 °C.

To prepare the derivatization reagent, 60 mg of ammonium iodide and 45 mg of dithiothreitol were dissolved in 30 mL of MSTFA and stored in the refrigerator at 4 °C for up to 2 months.

### 4.3. Urine Samples

Urine samples obtained from volunteers (males and females aged between 20 and 45) were used to construct calibration curves. The samples were preserved with sodium azide and stored at −20 °C before the analysis. To obtain a blank sample matrix for the optimization of SPAD conditions, urine samples were passed through the Bond Elute C18 (3 mL, 100 mg) (Agilent, Santa Clara, CA, USA) SPE cartridge, and the eluate was collected according to [35].

### 4.4. Instrumentation

A Thermo Trace 1310 gas chromatograph (Thermo Fisher Scientific, San Jose, CA, USA) coupled with a Thermo TSQ Quantum XLS mass spectrometer (Thermo Fisher Scientific, San Jose, CA, USA) operating with an electron ionization (EI) ion source was used. A capillary Thermo TG-1MT column (60 m × 0.25 mm × 0.25 µm) was used for the separation of analytes. Thermo XCalibur 2.2 software (Thermo Fisher Scientific, San Jose, CA, USA) was used for data collection and processing.

### 4.5. Liquid–liquid Extraction Procedure

A previously validated LLE procedure for steroids was used in a comparative study [41]. Briefly, 3 mL of urine was incubated in a 10 mL glass tube at 45 °C for 30 min with *E. coli* in a phosphate buffer solution (pH 6.5) to achieve full deconjugation of the analytes. Then, anhydrous sodium sulfate, 1 mL of carbonate buffer solution (pH 10), and 3 mL of diethyl ether were added for steroid extraction. After 3 min of extraction on a vortex mixer, the samples were centrifuged at 3000 rpm for 10 min. For phase separation, glass tubes were put into a cryostat thermostated at −35 °C to freeze the aqueous layer. The diethyl ether layer was transferred into another flask and evaporated in a dry block heater at 60 °C under a nitrogen stream. A 100 μL aliquot of the derivatization reagent was added to the dry residue, and the tube was incubated at 60 °C for 40 min. After cooling the tube to room temperature, 0.4 mL of acetonitrile was added, and the solution was analyzed by GC-MS/MS.

### 4.6. Optimum Solid-Phase Analytical Derivatization Conditions

Prior to the SPAD procedure, 2 mL urine samples were spiked with 0.6 mL of phosphate buffer (pH 6.5) containing β-glucuronidase from *E. coli* and thermostated for 30 min at 45 °C for deconjugation of the analytes. This volume of urine sample was selected according to the results confirming absence of breakthrough for the sample with volumes of up to 2 mL loaded on the cartridge.

Phenomenex Strata C18-E (100 mg, 1 mL) SPE cartridges were used in this study. These cartridges were preliminarily conditioned with 1 mL of acetonitrile and equilibrated with 1 mL of water. Then, the sample was loaded on the cartridge followed by a cartridge washing with 1 mL of a water–acetonitrile mixture (95:5, *v*/*v*). To prevent hydrolysis of the derivatization reagent, the cartridge was dried under a stream of nitrogen for 10 min. Then, 100 µL of the derivatization reagent was passed through the cartridge with its subsequent incubation in a dry block heater at 80 °C for 10 min. After cooling the cartridge to ambient temperature, 0.5 mL of acetonitrile was used to elute analytes.

## 5. Conclusions

A procedure for GC-MS/MS determination of TMS derivatives of steroid hormones using the SPAD methodology has been proposed, discussed, and compared with conventional methods. A comparison of the developed procedure with a conventional LLE has revealed that they have similar sensitivity, but SPAD provides higher recovery rates. Moreover, the complete sample preparation using SPAD requires less than an hour, while conventional procedures usually require more than 3 h, which significantly limits the throughput of the laboratory. Considering its simplicity and reproducibility, the proposed procedure can be used in doping control and clinical laboratories.

## Figures and Tables

**Figure 1 molecules-27-05796-f001:**
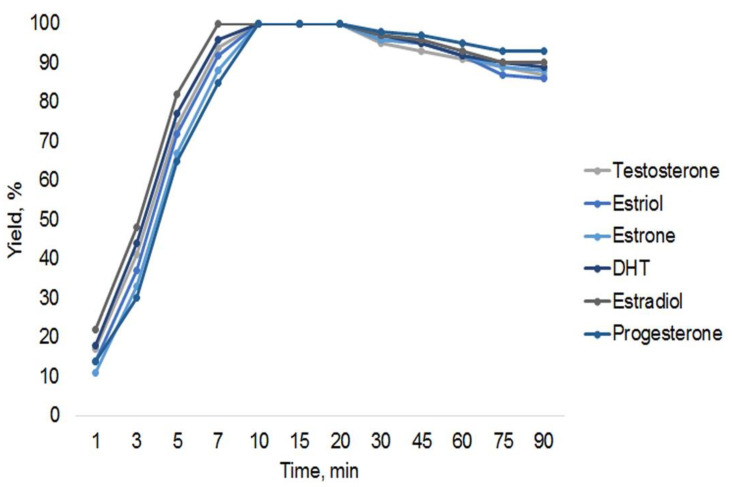
Influence of derivatization time on yield of derivatives obtained with SPAD sample preparation.

**Figure 2 molecules-27-05796-f002:**
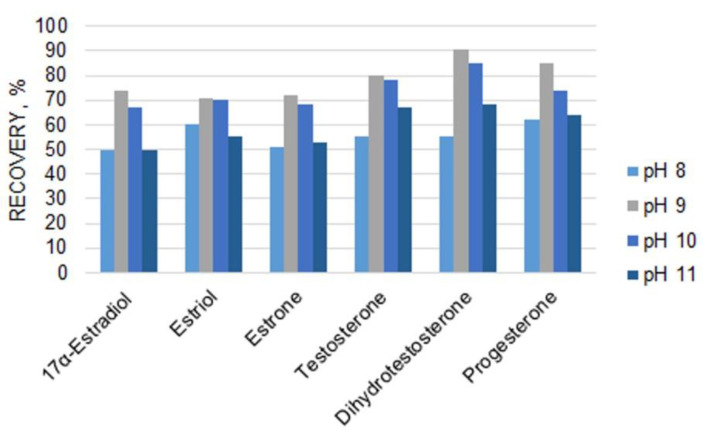
Influence of pH on steroid hormone recoveries for LLE with diethyl ether.

**Figure 3 molecules-27-05796-f003:**
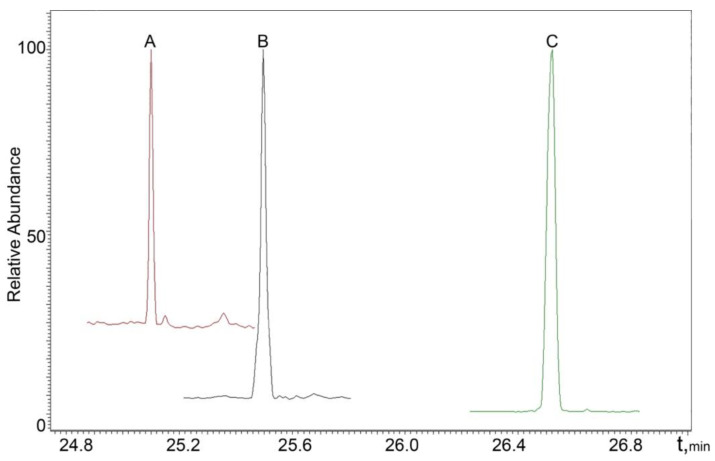
MRM chromatogram of a real male sample: A—DHT; B—testosterone; C—methyltestosterone (IS).

**Figure 4 molecules-27-05796-f004:**
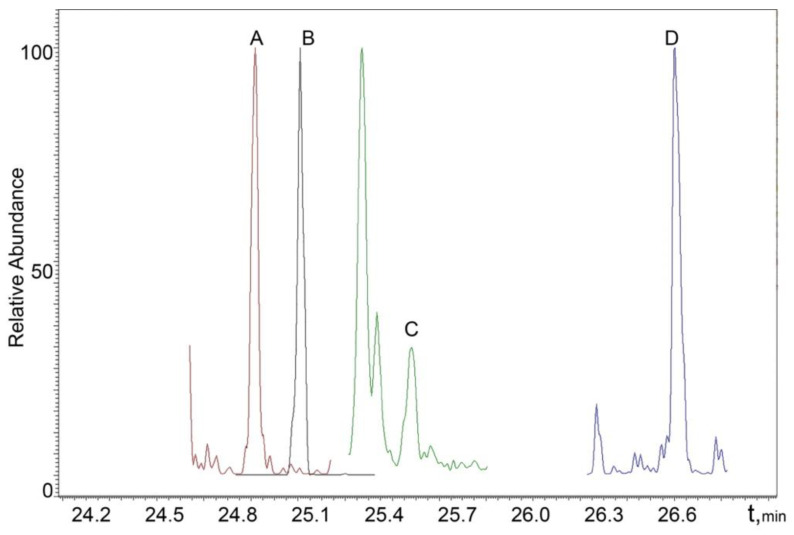
MRM chromatogram of a real female sample: A—17α-estradiol; B—estrone; C—testosterone; D—methyltestosterone (IS).

**Table 1 molecules-27-05796-t001:** An overview of some methods for the determination of steroids.

Analyte	Matrix	Sample Preparation	Method	LOQ (LOD), ng/mL	Reference
101 steroids	Tissue	Homogenization of 100 mg of the sample, centrifugation followed by lyophilization and dissolving in 150 µL of water–methanol (1:1) mixture	UHPLC-MS/MS (QqQ, ESI)	0.5–500	[34]
13 steroid glucuronides	Artificial urine	Enzymatic hydrolysis, SPE clean-up, evaporation, residue dissolution	UHPLC-MS/MS (QqQ, ESI)	1–500	[35]
7 steroids	Urine	Alkaline hydrolysis, LLE (MTBE), derivatization with MSTFA/DTT/NH_4_I mixture	GC-MS	1–4	[36]
13 steroids	Serum	LLE with ethyl acetate–n-hexane, incubation in ice for 10 min, centrifugation, evaporation, residue reconstruction	UHPLC-MS/MS (QqQ, ESI)	0.08–7.81	[37]
4 steroids	Urine	Protein precipitation with acetonitrile, SPE, evaporation and residue reconstruction	UHPLC-MS/MS (QqQ, ESI)	1.9–21.4 nmol/L	[38]

MTBE—methyl tert-butyl ether, DTT—dithiothreitol, MSTFA—N-methyl-N-(trimethylsilyl)trifluoroacetamide.

**Table 2 molecules-27-05796-t002:** Optimized MRM-transitions for steroid hormones determination.

Analyte	Precursor Ion, *m*/*z*	Product Ion, *m*/*z*	Collision Energy, eV	t_R_, min
17α-Estradiol	416.3	232.3	20	24.84
		284.8 *	10	
		326.8	5	
Estrone	414.3	231.2 *	30	25.03
		283.4	25	
		399.9	15	
Dihydrotestosterone	434.3	168.6	15	25.17
		195.3 *	30	
		291.2	20	
Testosterone	432.3	168.8	25	25.51
		403.8	15	
		417.5 *	10	
Methyltestosterone (IS)	446.4	195.0	30	26.53
		301.1 *	20	
		356.4	10	
Estriol	504.4	414.2	10	27.60
		311.1 *	15	
		255.0	30	
Progesterone	458.4	157.0	15	27.88
		208.1	25	
		443.4 *	10	

* Quantifier ion.

**Table 3 molecules-27-05796-t003:** Comparison of different methodologies for sample preparation used for steroid determination.

Procedure	Advantages	Disadvantages
Liquid–liquid extraction	Cost-effective; High solvent capacity; Easy to use	Requires toxic solvents; Time-consuming; pH-dependent character of extraction efficiency
Solid-phase extraction	Quick; Easy to use; Can be partially automated	Expensive; Requires toxic solvents; Sorbent capacity evaluation is required before analysis; Losses of analytes if solvent is evaporated after elution
Solid-phase analytical derivatization	Quick; Easy to use; Time-effective; Can be fully automated	Expensive; Sorbent capacity evaluation is required before analysis

**Table 4 molecules-27-05796-t004:** Analytical figures of merit of the proposed method.

Analyte	LOD, ng/mL	LOQ, ng/mL	Linear Range, ng/mL	R^2^
17α-Estradiol	2.5	5	5–100	0.995
Estrone	2.5	5	5–100	0.993
Dihydrotestosterone	1.0	2.5	2.5–100	0.996
Testosterone	1.0	2.5	2.5–100	0.997
Estriol	1.0	2.5	2.5–100	0.995
Progesterone	2.5	5	5–100	0.998

**Table 5 molecules-27-05796-t005:** Analytical performance of SPAD and LLE procedures combined with GC-MS/MS determination of steroid hormones.

Parameter	SPAD	LLE
Sensitivity (LOQ), ng/mL	2.5–5	1.0–2.5
Sample preparation time, min	60 (including 40 min of enzymatic hydrolysis)	180 (including 40 min of enzymatic hydrolysis)
Recovery, %	92–95	65–90

## Data Availability

Not applicable.

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
