# Peer review of "GC-MS/MS Determination of Steroid Hormones in Urine Using Solid-Phase Derivatization as an Alternative to Conventional Methods"

_molecules, 2022, doi:10.3390/molecules27185796_

Round 1

Reviewer 1 Report

Overview

In this excellent research work the authors employed solid-phase analytical derivatization (SPAD) technique which combines extraction and derivatization into a single step fulfilling many aspects of a good sample preparation technique, which includes low organic solvent consumption, economical, ease of automation with any chromatographic system and applicability in a wide range of complicated matrices. The authors derivatized steroid hormones using N-methyl-N-(trime- thylsilyl)trifluoroacetamide (MSTFA) as the derivatizing reagent.  The authors validated the SPAD method sufficiently and compared the proposed SPAD method with the more classical sample preparation method LLE in Table 5. I have only a few suggestions for the authors to improve the quality of this excellent manuscript. I have detailed below my comments.

1)     Line 231: Do not start a sentence with a number.

2)     Typically the term solid-phase is preferred over solid phase. Example solid-phase extraction and not solid phase extraction.  Please proofread carefully.

3)     Table 4. Analytical figures of merit of the proposed method: Any comment on reproducibility? RSD values if available?

Author Response

Reviewer 1:

Overview

In this excellent research work the authors employed solid-phase analytical derivatization (SPAD) technique which combines extraction and derivatization into a single step fulfilling many aspects of a good sample preparation technique, which includes low organic solvent consumption, economical, ease of automation with any chromatographic system and applicability in a wide range of complicated matrices. The authors derivatized steroid hormones using N-methyl-N-(trimethylsilyl)trifluoroacetamide (MSTFA) as the derivatizing reagent. The authors validated the SPAD method sufficiently and compared the proposed SPAD method with the more classical sample preparation method LLE in Table 5. I have only a few suggestions for the authors to improve the quality of this excellent manuscript. I have detailed below my comments.

Comment 1) Line 231: Do not start a sentence with a number.

Author response: Thank you for the comment! The sentence was revised.

Comment 2) Typically the term solid-phase is preferred over solid phase. Example solid-phase extraction and not solid phase extraction. Please proofread carefully.

Author response: Thank you for the comment! The corresponding changes were made in the manuscript.

Comment 3) Table 4. Analytical figures of merit of the proposed method: Any comment on reproducibility? RSD values if available?

Author response: Quality control solutions were not analyzed to assess the method reproducibility. However, the replicate analysis of calibration solutions which passed through the SPAD procedure (n = 6) showed that the RSD values were below 15% in the linear range of calibration curves.

Reviewer 2 Report

The submitted manuscript by Temerdashev et al. describes the development and evaluation of the analytical merits of the new GC-MS method for the quantification of six steroid hormones in urine. The study presentation and the experiment design must be improved before the manuscript con be accepted for publication.

Main remarks:

1.      It is unclear why the authors compare the simultaneous Solid Phase Extraction and Derivatisation sample preparation method with LLE, but not with SPE and subsequent derivatization method.

2.      In LLE the diethyl ether was evaporated under nitrogen flow? If not, it can be the reason for the reduced recovery for some steroids. What was the duration time of the evaporation?

3.      The authors should describe how the recoveries of the sample preparation methods were determined.          

4.      The analytes are eluted after 24 min in the time interval of 3 min. 24min is wasted time and analysis time should be reduced possibly by optimising the oven temperature program or by reducing column length.

5.      Sections 2.1 and 2.2 do not present the results but describe the methods and should be placed in the section Materials and methods.

6.      Lines 88-95. The text is for the introduction, not for the results section.

7. Table 2: the asterisks in the table are not explained 

Author Response

Reviewer 2:

Comments and Suggestions for Authors

The submitted manuscript by Temerdashev et al. describes the development and evaluation of the analytical merits of the new GC-MS method for the quantification of six steroid hormones in urine. The study presentation and the experiment design must be improved before the manuscript con be accepted for publication.

Main remarks:

Remark 1. It is unclear why the authors compare the simultaneous Solid Phase Extraction and Derivatisation sample preparation method with LLE, but not with SPE and subsequent derivatization method.

Author response: Thank you for the remark! LLE was used as a comparative procedure since this procedure is widely accepted in routine laboratories for the determination of steroid hormones in urine. At the same time, comparison of SPE and SPAD procedures under the same SPE conditions would only show the benefit of SPAD in terms of a decreased derivatization time in the cartridge compared to the solution.

Remark 2. In LLE the diethyl ether was evaporated under nitrogen flow? If not, it can be the reason for the reduced recovery for some steroids. What was the duration time of the evaporation?

Author response: Diethyl layer was evaporated to dryness by heating the tube in a dry block heater under nitrogen stream to prevent sample degradation. The proper remark was added to the manuscript in the section 4.5. Thermostating time of about 15 min was sufficient for complete solvent evaporation.

Remark 3. The authors should describe how the recoveries of the sample preparation methods were determined.

Author response: Thank you for the remark! The recoveries were calculated as the ratio of peak areas obtained by analyzing standard solutions of steroid hormone TMS-derivatives to those of solutions that passed through the sample preparation with equal final concentrations of the analytes (lines 162-164).

Remark 4. The analytes are eluted after 24 min in the time interval of 3 min. 24min is wasted time and analysis time should be reduced possibly by optimising the oven temperature program or by reducing column length.

Author response: Thank you for the remark! We are highly agreed that short columns (like 17 or 12 m columns with 0.1 mm ID) could be used for quick separation of most steroid hormones. It also makes GC separation time comparable with screening methods for UHPLC. The described GC separation conditions were selected due to their high efficiency expressed in the ability to separate not only analyte, but also isomer peaks in real samples. Indeed, the GC separation program could be slightly reduced; nevertheless, the GC-MS detection time was equal for LLE and SPAD procedures, which allowed to compare only sample preparation times.

Remark 5. Sections 2.1 and 2.2 do not present the results but describe the methods and should be placed in the section Materials and methods.

Author response: Thank you for the notice! Sections 2.1 and 2.2 were transferred to the “Materials and methods” section.

Remark 6. Lines 88-95. The text is for the introduction, not for the results section.

Author response: The text was transferred from lines 88-95 to the “Introduction” section (lines 40-45).

Remark 7. Table 2: the asterisks in the table are not explained

Author response: Thank you for the notice! The asterisk in Table 2 was explained.

Round 2

Reviewer 2 Report

All my remarks and questions have been answered and manuscript can be accepted for publication